# Isocaloric Substitution of Dietary Carbohydrate Intake with Fat Intake and MRI-Determined Total Volumes of Visceral, Subcutaneous and Hepatic Fat Content in Middle-Aged Adults

**DOI:** 10.3390/nu11051151

**Published:** 2019-05-23

**Authors:** Christa Meisinger, Susanne Rospleszcz, Elke Wintermeyer, Roberto Lorbeer, Barbara Thorand, Fabian Bamberg, Annette Peters, Christopher L. Schlett, Jakob Linseisen

**Affiliations:** 1Chair of Epidemiology, Ludwig-Maximilians Universität München at UNIKA-T Augsburg, 86156 Augsburg, Germany; j.linseisen@unika-t.de; 2Independent Research Group Clinical Epidemiology, Helmholtz Zentrum München, German Research Center for Environmental Health, 85764 Neuherberg, Germany; 3Institute of Epidemiology, Helmholtz Zentrum München, German Research Center for Environmental Health, 85764 Neuherberg, Germany; Susanne.rospleszcz@helmholtz-muenchen.de (S.R.); thorand@helmholtz-muenchen.de (B.T.); peters@helmholtz-muenchen.de (A.P.); 4Siegfried Weller Institute for Trauma Research, BG Trauma Center Tuebingen, Eberhard-Karls University Tuebingen, 72074 Tuebingen, Germany; EWintermeyer@bgu-tuebingen.de; 5Department of Radiology, Ludwig Maximilians University Hospital, 81377 Munich, Germany; Roberto.lorbeer@med.uni-muenchen.de; 6German Center for Diabetes Research (DZD), 85764 München-Neuherberg, Germany; 7Department of Diagnostic and Interventional Radiology, Faculty of Medicine, University of Freiburg, 79106 Freiburg, Germany; Fabian.bamberg@uniklinik-freiburg.de (F.B.); Christopher.schlett@uniklinik-freiburg.de (C.L.S.)

**Keywords:** hepatic fat content, visceral adipose tissue, body fat compartments, fat intake, diet, MRI

## Abstract

The present study investigated the association of carbohydrate intake and isocaloric substitution with different types of fat with visceral adipose tissue (VAT), subcutaneous adipose tissue (SAT) and hepatic fat content as determined by magnetic resonance imaging (MRI). Data from 283 participants (mean age 56.1 ± 9.0 years) from the MRI sub study of the KORA FF4 study were included. VAT, SAT and total body fat were quantified by a volume-interpolated VIBE-T1w-Dixon MR sequence. Hepatic fat content was determined as the proton density fat-fraction (PDFF) derived from multiecho-T1w MR sequence. Dietary intake was estimated using information provided by two different instruments, that is, repeated 24-h food lists and a food frequency questionnaire. Replacing total carbohydrates with an isoenergetic amount of total fat was significantly positively associated with VAT and hepatic fat, while there was no significant association with SAT. The multivariable adjusted β-coefficient for replacing 5% of total energy (5E%) carbohydrates with total fat was 0.42 L (95% CI: 0.04, 0.79) for VAT. A substitution in total fat intake by 5E% was associated with a significant increase in liver fat content by 23% (*p*-value 0.004). If reproduced in prospective studies, such findings would strongly argue for limiting dietary fat intake.

## 1. Introduction

Obesity, a worldwide epidemic due to the availability of many unhealthy food options and limited physical exercise [1], is a known risk factor for many metabolic disorders like coronary heart disease, malignancies, osteoarthritis and respiratory disorders [2].

Weight gain and a subsequent increase of body fat content causes an enlargement of adipocytes and goes along with an increase not only of subcutaneous fat but also of fat depots in other parts of the body. This ectopic fat is mainly stored intra-abdominally but also in muscle tissue, the liver and in the beta cells. Although both subcutaneous adipose tissue (SAT) and visceral adipose tissue (VAT) are metabolically active, the excess accumulation of VAT appears to play a more significant pathogenic role. Fat depots in liver and muscle tissue also can cause adverse cardiometabolic effects [3,4]. 

Between the late seventies and 2000, there was a change in the macronutrient composition of the diet; due to the advice on reducing fat intake, especially saturated fat, to decrease the incidence of dyslipoproteinemia and coronary heart disease (CHD), the relative proportion of energy from total fat decreased and that from carbohydrates increased over time [5]. In parallel, the contribution of carbohydrates from whole grain consumption to total carbohydrate intake decreased while the intake of refined starchy products and sugar-containing food increased [6,7].

The association between macronutrient intake and its influence on body fat depots was mainly investigated in animal studies [8] and very few human studies [9] so far. Here, we hypothesize that the proportion of carbohydrate and fat intake in the diet may play an important role in the accumulation of adipose tissue at different sites and liver fat accumulation in middle-aged adult men and women. Thus, the present study investigated the association of carbohydrate intake and isocaloric substitution with different types of fat with VAT, SAT and hepatic fat content as determined by MRI.

## 2. Materials and Methods

### 2.1. Study Population

The analysis was based on the KORA-FF4 study, the second follow-up study of the KORA Survey S4, conducted in 1999–2001. The study design was described in detail elsewhere [10]. Shortly, of those participating in the KORA-Survey S4, a total of 2279 men and women participated in the 14-year follow-up FF4 (2013/2014) [10]. A total of 400 individuals who participated in the FF4 study and met the inclusion criteria were included in a sub-study, the so-called KORA-MRI study. All subjects included in this sample, which was enriched with individuals with type 2 diabetes and prediabetes, received a whole body MRI. The study design, the sampling procedure, the data collection and the inclusion and exclusion criteria were described in detail elsewhere [11]. For the present analysis, a total of 117 study participants had to be excluded because of missing values in the MRI parameters of interest due to insufficient image quality, imaging artifacts or technical errors (*n* = 29), missing nutrition data (*n* = 82) or missing values in any covariable (*n* = 6) (compare Figure 1). 

### 2.2. Dietary Assessment

Dietary intake was assessed using a self-administered food frequency questionnaire (FFQ) and repeated 24-h food lists [12]. Combining this information, the usual dietary intake was estimated in a two-step approach. The consumption probability and the consumption amount on consumption days were estimated separately with models both including the same covariates to link the two parts. Then, the usual dietary intake of all food items was calculated for each participant by multiplying the consumption probability of a certain food item by the usual consumption amount on a consumption day [13,14]. Nutrient contents of food items were derived from the German Nutrient Database Bundeslebensmittelschlüssel, version 3.02. For the present analysis, we focused on energy-providing nutrients (fat, carbohydrates, protein, alcohol) and the fat subgroups of saturated fatty acids (SFA), monounsaturated fatty acids (MUFA) and polyunsaturated fatty acids (PUFA). Alcohol intake was also assessed and categorized into three categories: no or very low alcohol consumption (<2 g/day for women, <5 g/day for men); moderate alcohol intake (≥2 g/day to <10 g/day for women; ≥5 g/day to <20 g/day for men) and high alcohol intake (≥10 g/day for women, ≥20 g/day for men).

### 2.3. Magnetic Resonance Imaging (MRI)

MRI scans were performed with a 3 Tesla Magnetom Skyra (Siemens Healthineers, Erlangen, Germany) equipped with a whole-body coiling system. Details on the whole-body protocol are described in detail elsewhere [11]. A multiecho Dixon-sequence was used to quantify the lipid content at the level of the portal vein [15]. The hepatic adipose content is given in %.

SAT and VAT was measured by a 3D in/opposed-phase VIBE Dixon sequence and segmented by an automated procedure based on fuzzy-clustering. SAT in liter (L) was quantified from the diaphragm to the femoral head and VAT in L was measured from the cardiac apex to the femoral head [4]. 

### 2.4. Assessment of Covariables

After an overnight fast of at least 8 h, all non-diabetic participants underwent a standard 75-g oral glucose tolerance test (OGTT). Newly diagnosed diabetes (NDD; ≥126 mg/dL fasting plasma glucose or ≥200 mg/dL 2-h post glucose load), i-IFG (fasting plasma glucose ≥110 mg/dL but <126 mg/dL and 2-h post glucose load <140 mg/dL), i-IGT (fasting plasma glucose <110 mg/dL and 2-h post glucose load ≥140 mg/dL but <200 mg/dL), IFG/IGT and normal glucose tolerance (NGT; fasting plasma glucose <110 mg/dL and 2-h post glucose load <140 mg/dL) were defined according to the 1999 WHO diagnostic criteria as described elsewhere [16].

Information on sociodemographic characteristics, lifestyle and risk factors and medications was gathered during a standardized face-to-face interview. In addition, study participants underwent a medical examination as described in more detail elsewhere [17]. BMI was calculated as weight in kilograms divided by the square of height in meters. Systolic and diastolic blood pressure were measured three times using the right arm of seated participants, after at least 5 min at rest. Subsequently, the mean of the second and third measurement was calculated. Hypertension was defined as blood pressure values ≥140/90 mmHg and/or the use of antihypertensive medication, given that the individuals were aware of being hypertensive. Physical activity assessment was based on information regarding leisure-time physical training during summer and winter and a 4 categories variable was built [18]. 

### 2.5. Clinical Chemical Measurements

A fasting venous blood sample was obtained from all study participants while sitting. All parameters were measured immediately. Blood glucose was analyzed using an enzymatic reference method with hexokinase. HbA1c was determined using a cation-exchange high performance liquid chromatographic, photometric VARIANT II TURBO HbA1c Kit (Bio-Rad Laboratories Inc., Hercules, CA, USA). 

Total serum cholesterol, HDL-cholesterol, LDL-cholesterol and triglycerides analyses were carried out using an enzymatic, colorimetric method. Liver enzymes were measured using an enzymatic, colorimetric method. After about half of the study period, the KORA FF4 measurement instrument and assays changed from Siemens to Roche. Calibration formulas were developed using 122 KORA FF4 samples, which were measured with both instruments/assays during the time of the method change. The Siemens measurement results were calibrated to correspond to the Roche measurements.

### 2.6. Statistical Analysis

The final sample size comprised *n* = 283 participants. Sociodemographics and other covariates, nutrition variables and MRI adipose tissue outcomes are presented as arithmetic mean and standard deviation or as median with first and third quartile for hepatic fat content. Differences between men and women were determined by t-test, Wilcoxon Rank Sum Test and χ^2^-Test, as appropriate.

Daily carbohydrate, fat and protein intake was transformed as % of total energy intake by assuming a mean energy value of 9 kcal/g for fat and fat subgroups, 4 kcal/g for carbohydrates and protein and 7 kcal/g for alcohol. Correlations between dietary intake and MRI adipose tissue outcomes were determined by Pearson correlation coefficients and corresponding *p*-values.

To assess the association of the MRI derived adipose tissue parameters as outcomes with an isocaloric replacement of carbohydrates by fat or fat subtypes as exposure, a substitution model based on linear regression was calculated. The substitution model contains total energy intake and all energy-providing nutrients except carbohydrates (i.e., fat, protein and alcohol) and was additionally adjusted for potential confounding covariates. Variables for fat, protein and alcohol were adequately scaled so that β-coefficients for energy-providing nutrients denote a change per 5% of total energy intake (5E%). Thus, the estimated coefficient for fat can be interpreted as the association of the adipose tissue parameter with a 5E% increase in fat at the expense of carbohydrates while energy supply from protein and alcohol remains unchanged. As potentially confounding variables, age, sex and glycemic status were included in the model. Hepatic fat content was log-transformed before analysis and therefore estimates have to be interpreted as percent change. We repeated the analyses including only persons with normal glucose tolerance (*n* = 183; sensitivity analysis).

R version 3.4.4 was used for all calculations. *p*-values < 0.05 are considered to denote statistical significance.

## 3. Results

The characteristics of the participating men and women as well as for the total sample are presented in Table 1. The mean age of the entire study cohort was 56.1 (±9.0) years. In comparison to women, men reported more years of schooling; males had a higher waist circumference, higher waist-to-hip-ratio, higher systolic and diastolic blood pressure values, higher LDL-cholesterol and triglycerides values and lower HDL-cholesterol values. Furthermore, higher values regarding the liver enzymes GGT, GOT and GPT were seen in male participants in comparison to females. Also, in men prediabetes and diabetes were more prevalent than in women. They were more often physically active, less often never-smoker and reported less often no or very low alcohol intake. 

In Table 2, dietary intake data and the results of the MRI-measurements of adipose tissue for the total sample and separately for men and women are given. In the total sample, mean energy intake by carbohydrate and fat intake was 41.8 E% and 37.9 E%, respectively. Men had higher volumes of total adipose tissue and VAT and higher hepatic fat content compared with women, while women had a higher volume of subcutaneous adipose tissue than men.

In Pearson correlation analysis (Figure 2, Figure 3 and Figure 4), carbohydrate intake showed an inverse correlation with VAT and hepatic fat content, while total fat intake was significantly positively correlated with SAT and hepatic fat. Considering the fat subtypes, MUFAs were significantly positively correlated with all three fat compartments, while PUFAs correlated positively with SAT only and SFAs showed no significant correlations with VAT, SAT or hepatic fat content.

Replacing total carbohydrates with an isoenergetic amount of total fat was significantly positively associated with VAT and hepatic fat, while there was no significant association with SAT (Table 3). The multivariable adjusted β-coefficient for increasing fat intake by 5E% at the expense of carbohydrate intake was 0.42 L [95% CI: 0.04, 0.79] for VAT. An increase in total fat intake by 5E% at the expense of carbohydrate was associated with an increase in liver fat by 23%. Substitution of carbohydrates by SFAs or MUFAs was not significantly associated with VAT and hepatic fat content but with SAT; substitution by PUFAs was not associated with any of the fat compartments.

In a sensitivity analysis including only subjects with normal glucose tolerance status stronger effects for fat and MUFA were found. Altogether, the correlations pointed in the same directions as those identified in the main analysis. Furthermore, in the substitution models (Table 3) comparable results could be generated.

## 4. Discussion

This study is the first to demonstrate the association between macronutrient intake and different fat depots measured by MRI, which provided accurate information on the quantity of fat tissue and liver fat accumulation for each study participant. We could show that isocaloric substitution of total carbohydrates with an isoenergetic amount of total fat was substantially positively associated with VAT and hepatic fat, while there was no significant association with SAT. Substitution of carbohydrates by PUFAs was not significantly associated with any of the fat compartments while a substitution by SFAs or MUFAs was significantly associated with SAT but not VAT and hepatic fat content.

The findings on isocaloric substitution of carbohydrates by SFAs are in line with controlled intervention studies, which reported that an increase of dietary SFA in isocaloric substitution of carbohydrates [19] or PUFA [20] increased hepatic and—contrary to our results—visceral fat accumulation in healthy subjects. Prior cross-sectional studies found that persons with higher intake of SFA have increased liver fat content, determined by liver enzymes measurement [21,22]. In general, the associations with hepatic fat seems to point in the same direction, although our results were not statistically significant which is probably due to the low number of persons included.

Isocaloric substitution of carbohydrates with total fat was significantly positively associated with VAT, hepatic fat but not SAT in the present study. To the best of our knowledge, no other studies investigated this association. However, due to prior work it can be assumed that an increase in the intake of total fat and SFA may promote visceral fat accumulation [20,23] and non-alcoholic fatty liver disease [24]. Furthermore, in several prior intervention studies, high-fat diets resulted in increased liver fat accumulation in both humans and rodents when compared with low-fat diets [19,25,26]. On the other hand, a prior cross-sectional study found an association between dietary fat and weight in adult women without diabetes but could not show a relationship with VAT measured by dual-energy X-ray absorptiometry [27]. The different findings in the studies could be due to methodological differences in assessing diet and measurement of the fat compartments. For example, in the study by Greenfield et al. [27] diet was assessed using the Oxford FFQ and VAT was measured using dual-energy X-ray absorptiometry, while in the present study dietary fat intake was assessed using a blended approach leading to more precise and valid intake estimates as compared to FFQ-based intake data [28] and VAT was determined by MRI.

Contrary to the results of the substitution models, we found no significant correlation between total fat intake and VAT but with SAT. As total fat is the sum of different fat types, it can be assumed that the consumption of certain fat subtypes might be responsible for the different results. Indeed, our study showed a strong positive association between the intake of MUFAs and PUFAs (more than 50% of the total dietary fat) but not SFAs and SAT. This likely explains the positive correlation between total fat intake and SAT.

Independent of the caloric content, high-fat diets up-regulate inflammatory mediators [29], promote unfavorable epigenetic profiles [30,31], stimulate hepatic bile acid synthesis and subsequently may promote tumorigenesis [32,33]. There is also evidence that the type of fat consumed plays an important role in this context; isocaloric diets with a reduced SFA or increased MUFA or PUFA content, appear to reduce liver fat [34]. Main food sources of SFAs are foods of animal origin high in fat such as cream, cheese, butter, other whole milk dairy products and fatty meats; however, also some vegetables fats are rich in SFA, for example, coconut and palm kernel oils. A prior German study investigating whether and to what extent patterns of nutrient intake are associated with VAT and subcutaneous abdominal adipose tissue (SAAT) determined by MRI observed that VAT was primarily explained by nutrient quality, while SAAT was explained by total energy intake [9]. In that study nutrients found in animal (except for dairy) products were associated with VAT [9]; but unfortunately, the association between macronutrient composition and different fat depots using substitution modelling was not investigated and thus, the results are not comparable to our findings [9].

Carbohydrate consumption correlated negatively with VAT and hepatic fat content in our study. This finding is contrary to the results of studies reporting that diets higher in refined carbohydrates possibly are accompanied by an increase of liver fat [35]. The different findings are probably due to differences in the quality of consumed carbohydrates. Ideally, the food supplied should contain low-glycemic carbohydrates such as minimally processed grains, legumes and non-starchy fruits and vegetables [36]. The benefit through the consumption of carbohydrates with a low-glycemic index was underpinned by a recent RCT [37].

### Differences between VAT and SAT

There are many differences between VAT and SAT. Compared to SAT, VAT adipocytes have a higher rate of lipolysis, which is more readily stimulated by catecholamines and less readily stimulated by insulin [38]. VAT is metabolically active and produces more than 20 different hormones, such as the stress hormone cortisol, leptin and adiponectin (responsible for regulating satiety and food intake) [39]. Subsequently, insulin resistance and many of its related features could arise from VAT [40,41]. Furthermore, in contrast to the SAT, the VAT is anatomically linked to the liver via portal vein. Thus, VAT can deliver free fatty acids and adipokines/cytokines directly to the liver, which may explain the relationship between VAT and an increased content of liver fat [42]. Preadipocytes from SAT depots have a greater capacity than VAT to differentiate into numerous, small, insulin-sensitive, adipocytes, which act as lipid-storing cells [43]. SAT can be considered as the normal physiological buffer for excess energy intake in form of a hypercaloric diet in combination with limited energy expenditure via physical activity. Adipocytes in SAT store excess free fatty acids and glycerol in form of triglycerides [44] and when the capacity to store excess energy and the ability to generate new adipocytes is impaired, fat is stored outside the SAT [45]. Altogether, the anatomical distribution of adipose tissue shows a subject-to-subject variation, dependent on age, sex, ethnicity, nutritional intake and the autonomic regulation of energy homeostasis [46,47] but the mechanisms involved in determination of fatness and fat distribution are barely understood so far. Future research should focus on the molecular and cellular mechanisms in different adipose tissue depots and on adipocyte-specific factors to better understand how the growth and turnover of visceral versus subcutaneous adipocytes is regulated.

The limitations of the study are the relatively low number of participants and the enrichment of the study sample with persons with prediabetes and diabetes. However, when conducting sensitivity analyses including normoglycemic men and women only, the results pointed in the same direction as those found in the main analyses. Furthermore, our results are not generalizable to other populations and ethnicities or other age-groups because only Caucasians with a mean age of 56.1 years were included. In addition, the cross-sectional design of the present study did not allow to postulate a causal link between macronutrient intake and fat enrichment in different body compartments. The major strengths of our study are the well-characterized participants and the availability of data on lifestyle and multiple risk factors, measured following standardized protocols. A further important strength is the accurate measurement of adipose tissue compartments using an MRI for the quantification of fat.

In conclusion, using MRI and a relatively accurate dietary assessment method compared to other methods [48], we observed an association between fat accumulation at specific anatomic locations and macronutrient composition among adult German men and women. Specifically, the isocaloric substitution of carbohydrates with fat was associated with higher hepatic fat content and visceral fat accumulation. These findings provide the basis for future studies that investigate the effect of diet and on metabolic and molecular consequences of regional obesity. Also, the results can contribute to the long-lasting discussion on the diet’s optimal fat content.

## Figures and Tables

**Figure 1 nutrients-11-01151-f001:**
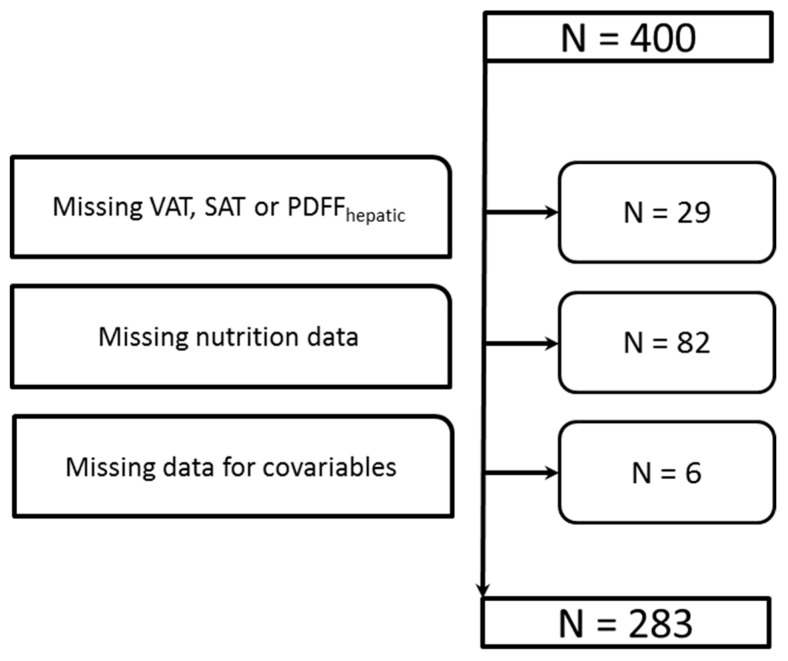
After excluding subjects with missing values, a total of 283 participants who underwent whole-body magnetic resonance imaging (MRI) could be included in the analysis.

**Figure 2 nutrients-11-01151-f002:**
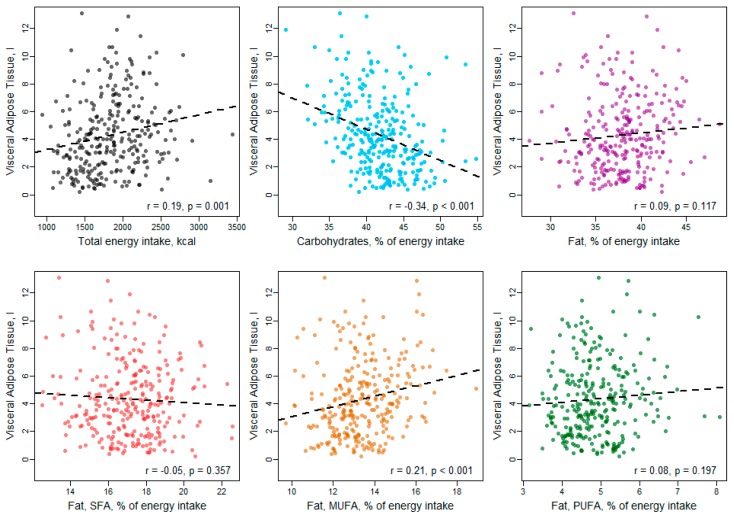
Correlation between macronutrient intake and visceral adipose tissue.

**Figure 3 nutrients-11-01151-f003:**
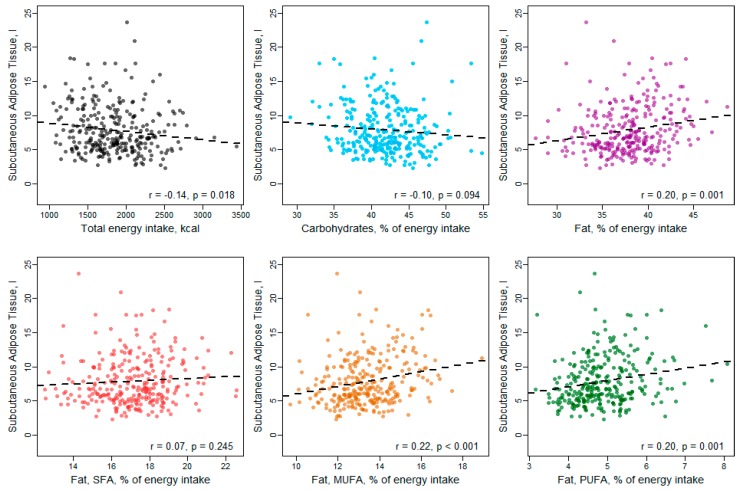
Correlation between macronutrient intake and subcutaneous adipose tissue.

**Figure 4 nutrients-11-01151-f004:**
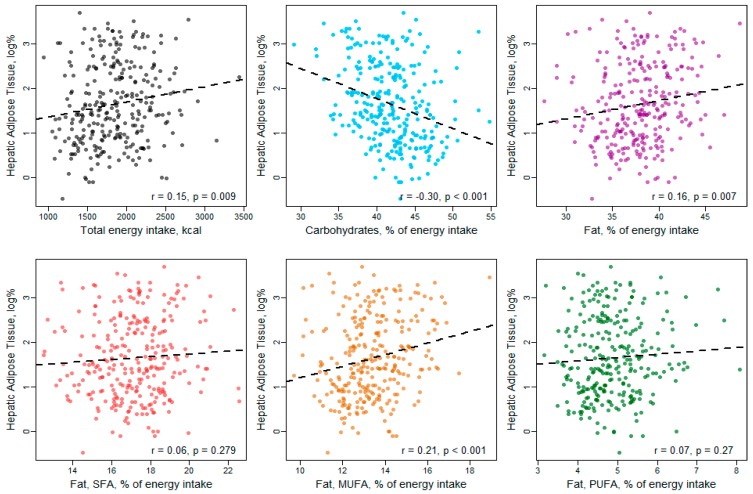
Correlation between macronutrient intake and hepatic fat content.

**Table 1 nutrients-11-01151-t001:** Baseline characteristics of the whole study sample as well as for men and women (covariables only).

	All	Men	Women	
	*N* = 283	*N* = 161	*N* = 122	*p*-value Men vs. Women
**Sociodemographics**				
Age, years	56.1 ± 9.0	56.1 ± 9.4	56.1 ± 8.6	0.976
Education, years of schooling	12.4 ± 2.6	12.9 ± 2.8	11.8 ± 2.4	0.001
**Anthropometric measurements**				
Height, cm	171.7 ± 9.7	177.9 ± 6.7	163.5 ± 6.4	<0.001
Weight, kg	81.4 ± 15.7	88.4 ± 13.1	72.2 ± 14.1	<0.001
BMI, kg/m^2^	27.6 ± 4.6	27.9 ± 4.1	27.0 ± 5.2	0.107
Waist circumference, cm	97.0 ± 13.7	102.2 ± 11.7	90.2 ± 13.3	<0.001
Hip circumference, cm	106.1 ± 8.3	106.4 ± 6.9	105.7 ± 9.9	0.483
Waist-To-Hip Ratio	0.9 ± 0.1	1.0 ± 0.1	0.9 ± 0.1	<0.001
**Blood Pressure**				
Systolic BP, mmHg	119.9 ± 15.9	125.1 ± 15.4	113.0 ± 13.9	<0.001
Diastolic BP, mmHg	74.8 ± 9.6	76.9 ± 10.0	71.9 ± 8.4	<0.001
Hypertension	94 (33.2%)	64 (39.8%)	30 (24.6%)	0.011
Antihypertensive medication	71 (25.1%)	42 (26.1%)	29 (23.8%)	0.759
**Diabetes**				
Glycemic Status				0.018
normoglycemic	183 (64.7%)	93 (57.8%)	90 (73.8%)	
prediabetes	68 (24.0%)	45 (28.0%)	23 (18.9%)	
diabetes	32 (11.3%)	23 (14.3%)	9 (7.4%)	
Fasting Serum glucose, mg/dL	102.8 ± 18.0	106.6 ± 19.5	97.8 ± 14.3	<0.001
HbA1c, %	5.5 ± 0.6	5.5 ± 0.6	5.5 ± 0.5	0.671
Diabetes medication	20 (7.1%)	13 (8.1%)	7 (5.7%)	0.599
**Lipid profile**				
Total Cholesterol, mg/dL	217.2 ± 36.6	216.1 ± 37.8	218.6 ± 35.0	0.577
HDL Cholesterol, mg/dL	62.8 ± 18.1	56.0 ± 14.9	71.8 ± 17.9	<0.001
LDL Cholesterol, mg/dL	138.8 ± 33.6	141.7 ± 34.1	135.0 ± 32.6	0.099
Triglycerides, mg/dL	127.2 ± 80.4	148.1 ± 95.0	99.5 ± 42.1	<0.001
Lipid lowering medication	31 (11.0%)	16 (9.9%)	15 (12.3%)	0.662
**Liver enzymes**				
GGT - Gammaglutamyltransferase, µkat/L	0.7 ± 0.7	0.8 ± 0.8	0.5 ± 0.5	<0.001
Glutamat-Oxalat-Transaminase (GOT, AST), µkat/L	0.4 ± 0.2	0.4 ± 0.2	0.4 ± 0.2	0.008
Glutamat-Pyruvat-Transaminase (GPT, ALT), µkat/L	0.5 ± 0.3	0.6 ± 0.3	0.4 ± 0.3	<0.001
**Behavior**				
Physical activity				
no	69 (24.4%)	49 (30.4%)	20 (16.4%)	0.041
sporadically	41 (14.5%)	22 (13.7%)	19 (15.6%)	
regularly around 1 h/week	92 (32.5%)	45 (28.0%)	47 (38.5%)	
regularly more than 2 h/week	81 (28.6%)	45 (28.0%)	36 (29.5%)	
Smoking				0.249
never-smoker	105 (37.1%)	54 (33.5%)	51 (41.8%)	
ex-smoker	122 (43.1%)	76 (47.2%)	46 (37.7%)	
smoker	56 (19.8%)	31 (19.3%)	25 (20.5%)	
Alcohol consumption *				
No/very low	72 (25.4%)	31 (19.3%)	41 (33.6%)	<0.001
Moderate	133 (47.0%)	65 (40.4%)	68 (55.7%)	
high	78 (27.6%)	65 (40.4%)	13 (10.7%)	

Values are arithmetic means and standard deviation for continuous variables and counts and percentages for categorical variables. *p*-values from *t*-test or χ^2^-Test, where appropriate; * alcohol consumption: no or very low (<2 g/day for women, <5 g/day for men), moderate (≥2 g/day to <10 g/day for women, ≥5 g/day to <20 g/day for men) and high alcohol intake (≥10 g/day for women, ≥20 g/day for men).

**Table 2 nutrients-11-01151-t002:** Dietary intake as well as MRI variables in the KORA MRI sub study, by sex.

	All	Men	Women	
	*N* = 283	*N* = 161	*N* = 122	*p*-value Men vs. Women
**Dietary intake data**				
Carbohydrates, % of total energy intake	41.8 ± 4.0	41.3 ± 4.3	42.5 ± 3.5	0.015
Fat, % of total energy intake	37.9 ± 3.5	37.3 ± 3.5	38.7 ± 3.3	0.001
Ratio fat/carbohydrates	0.9 ± 0.2	0.9 ± 0.2	0.9 ± 0.1	0.895
SFA, % of total energy intake	17.1 ± 1.8	16.8 ± 1.8	17.6 ± 1.8	0.000
MUFA, % of total energy intake	13.4 ± 1.5	13.4 ± 1.5	13.5 ± 1.4	0.562
PUFA, % of total energy intake	4.9 ± 0.8	4.8 ± 0.8	5.0 ± 0.7	0.002
Protein, % of total energy intake	15.3 ± 1.7	14.8 ± 1.5	15.8 ± 1.7	<0.001
Alcohol, % of total energy intake	4.3 ± 3.7	5.9 ± 3.7	2.2 ± 2.3	<0.001
Total energy intake, kcal		2061.5 ± 351.5	1554.8 ± 295.4	<0.001
**MRI measurements of adipose tissue**				
Hepatic fat, PDFF, % (median [1st quartile, 3rd quartile])	4.5 [2.5, 11.1]	6.6 [3.7, 12.8]	3.0 [1.9, 5.4]	<0.001
Total adipose tissue, liter	12.2 ± 5.4	12.8 ± 5.2	11.5 ± 5.5	0.037
Visceral adipose tissue, liter	4.3 ± 2.7	5.5 ± 2.6	2.8 ± 2.1	<0.001
subcutaneous adipose tissue, liter	7.9 ± 3.6	7.3 ± 3.1	8.7 ± 3.9	0.001

Values are arithmetic means and standard deviation, unless otherwise indicated. *p*-values from *t*-test or Wilcoxon Test, where appropriate. SFA, saturated fatty acids; MUFA, monounsaturated fatty acids; PUFA, polyunsaturated fatty acids.

**Table 3 nutrients-11-01151-t003:** Effects of substitution of carbohydrates by total fat, saturated fatty acids (SFA), monounsaturated fatty acids (MUFA) and polyunsaturated fatty acids (PUFA) on visceral adipose tissue (VAT), subcutaneous adipose tissue (SAT) and liver fat content in the KORA MRI sub study.

	VAT			SAT			Hepatic		
	β-Coefficient	95%-CI	*p*-Value	β-Coefficient	95%-CI	*p*-Value	Estimate	95%-CI	*p*-Value
Fat	0.42	[0.04, 0.79]	**0.031**	0.15	[−0.47, 0.76]	0.642	1.23	[1.07, 1.42]	**0.004**
Fat *	0.63	[0.21, 1.05]	**0.003**	0.19	[−0.52, 0.91]	0.595	1.2	[1.01, 1.42]	**0.034**
SFA	−0.04	[−0.95, 0.86]	0.924	−1.65	[−3.11, −0.19]	**0.027**	1.3	[0.92, 1.80]	0.138
SFA *	0	[−0.99, 0.98]	0.998	−1.34	[−3.02, 0.33]	0.115	1.26	[0.85, 1.88]	0.247
MUFA	0.98	[−0.32, 2.27]	0.138	2.58	[0.49, 4.68]	**0.016**	1.23	[0.76, 1.99]	0.399
MUFA *	1.89	[0.36, 3.42]	**0.016**	2.58	[−0.03, 5.19]	0.052	1.14	[0.61, 2.12]	0.680
PUFA	0.13	[−2.00, 2.26]	0.905	−1.81	[−5.25, 1.63]	0.302	1.01	[0.45, 2.25]	0.979
PUFA *	−1.12	[−3.69, 1.44]	0.388	−2.22	[−6.58, 2.14]	0.317	1.09	[0.39, 3.06]	0.859

Substitution models contained total energy intake, protein intake, alcohol intake and fat (subtype) intake. Estimates are therefore interpreted as the association of the adipose tissue parameter with a 5 E% increase in fat at the expense of carbohydrates while energy supply from protein and alcohol remains unchanged. Models were additionally adjusted for age, sex and glycemic status to avoid potential confounding by these variables. In the analysis of fat subtypes (SFA, MUFA and PUFA), adjustments were made for the other fat subtypes (instead of total fat); * Results from the sensitivity analysis including only persons with normal glucose tolerance. Bold: *p*-values denote significant results.

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
