# Peer review of "Isocaloric Substitution of Dietary Carbohydrate Intake with Fat Intake and MRI-Determined Total Volumes of Visceral, Subcutaneous and Hepatic Fat Content in Middle-Aged Adults"

_nutrients, 2019, doi:10.3390/nu11051151_

Reviewer 1 Report

Meisinger et al. investigated the association of carbohydrate intake and isocaloric substitution with different types of fat with VAT, SAT, and hepatic fat content as determined by MRI. The study is elegant, original and well written. The experimental section is descriptive with the detailed discussion of the results. The manuscript has a good amount of citations to the references.  

Only one minor concern: Text format is not consistent. Please check out.

Author Response

Response to Reviewer 1

We thank the reviewer very much for the positive review of our manuscript. We made changes to the manuscript as suggested.

Text format is not consistent. Please check out.

Answer: As suggested, the text format of the manuscript is consistent now.

Reviewer 2 Report

In the present study, the authors want to investigate the association of carbohydrate intake and isocaloric substitution with different types of fat with visceral adipose tissue (VAT), subcutaneous adipose tissue (SAT), and hepatic fat content as determined by MRI.

The authors observed an association between fat accumulation at specific anatomic locations and macronutrient composition among adult German men and women. Specifically, the isocaloric substitution of carbohydrates with fat was associated with higher hepatic fat content and visceral fat accumulation.

The findings are related to a specific range of age Caucasians with a mean age of 56.1 years, but it provides the basis for future studies that investigate the effect of diet and on metabolic and molecular consequences of regional obesity.

The introduction provides sufficient background and includes relevant references. The research design is appropriate and the conclusions are enough reported by the results.

References are updated

My recommendation is to accept after minor revisions related to

·        The authors should explain with more detail the statistical analysis.

They should remind the 5E% model, and the adjustment of the model for age, sex, and glycemic status.

·         Ref 10 should be revised for text editing: 2007 instead of 20007

Author Response

Response to Reviewer 2

We thank the reviewer very much for the positive review of our manuscript. We have revised the paper accordingly.

The authors should explain with more detail the statistical analysis.

They should remind the 5E% model, and the adjustment of the model for age, sex, and glycemic status.

Answer: As suggested by the reviewer, we now explain the statistical analysis in more detail (see page 4 last paragraph and page 5 first line). Furthermore, we now describe the results presented in Table 3 in more detail to make the 5E% model, and the adjustment of the model for age, sex, and glycemic status more clear (see page 9).

Ref 10 should be revised for text editing: 2007 instead of 20007

Answer: Thank you for this hint. We have omitted this mistake (see page 12).